# A Quasi-Experimental Hip-Hop-Based Program to Improve Motor Competence and Physical Activity in Preschoolers in Portugal: The “Grow+” Program

**DOI:** 10.3390/healthcare13192518

**Published:** 2025-10-04

**Authors:** Cristiana Mercê, Sofia Bernardino, Neuza Saramago, Marco Branco, David Catela

**Affiliations:** 1Sport Sciences School of Rio Maior (ESDRM), Santarém Polytechnic University, Av. Dr. Mário Soares, 110, 2040-413 Rio Maior, Portugal; sofia.bernardino@hotmail.com (S.B.); neuzasaramagorocha@gmail.com (N.S.); marcobranco@esdrm.ipsantarem.pt (M.B.); catela@esdrm.ipsantarem.pt (D.C.); 2Sport Physical Activity and Health Research & Innovation (SPRINT), Santarém Polytechnic University, Complex Andaluz, Apart 279, 2001-904 Santarém, Portugal; 3Physical Activity and Health—Life Quality Research Centre (CIEQV), Polytechnique University of Santarém, Complex Andaluz, Apart 279, 2001-904 Santarém, Portugal; 4Interdisciplinary Center for the Study of Human Performance (CIPER), Faculty of Human Kinetics, University of Lisbon, Cruz Quebrada-Dafundo, 1499-002 Lisboa, Portugal; 5Quality Education—Life Quality Research Centre (CIEQV), Santarém Polytechnique University, Complex Andaluz, Apart 279, 2001-904 Santarém, Portugal

**Keywords:** hip hop, dance, motor development, early childhood, motor skills, physical activity, intervention study, health

## Abstract

**Background/Objectives**: Dance, particularly hip-hop, offers a dynamic means of fostering physical activity (PA) and encouraging movement in health-related initiatives among children and youth in educational environments. Hip-hop offers benefits across motor, physical, social, and mental domains. Given the importance of PA in early development, and the preschool period as a sensitive phase for acquiring motor skills, this study aimed to examine the effects of the “Grow+” hip-hop program on motor competence (MC), perceived motor coordination (PMCoor), and PA levels in preschoolers. **Methods**: A quasi-experimental within-subjects design was used, including 37 children aged 3 to 4 (M = 4.29 ± 0.58). The intervention included two 4-week hip-hop periods, separated by a 4-week break. Four assessments were conducted using the MCA battery (MC), PA’s pictorial scales, and questionnaires completed by caregivers and educators (PMCoor). Data were analyzed using repeated measures ANOVA and Spearman correlations. **Results**: MC and PA levels showed a nonsignificant but positive trend across the study. Significant improvements in MC were observed during intervention periods, while no significant changes occurred during the break. Educators’ perceptions of PMCoor remained unchanged, despite improvements in MC. **Conclusions**: The findings suggest that the “Grow+” hip-hop program contributed meaningfully to improvements in MC and PA levels among children in early childhood. These findings accentuate the potential efficacy of structured rhythmic movement interventions in promoting motor development throughout early childhood, thereby supporting their integration into early childhood education curricula.

## 1. Introduction

Engaging in physical activity (PA) is widely recognized as a salutogenic behavior that promotes numerous lifelong health benefits, including enhanced cardiorespiratory fitness, muscular strength, and well-being in children with both typical and atypical patterns of motor development [1,2,3,4,5]. PA is defined as any bodily movement produced by the contraction of skeletal muscles that increases caloric demand, compared to resting energy expenditure [6]. Motor competence (MC) also plays an important role in children’s growth, development, and their likelihood of adopting an active lifestyle. This concept is often specified as proficiency in performing fundamental motor skills (FMSs) (e.g., throwing, catching, running, jumping) [7], which are ideally acquired during early and middle childhood and are typically categorized into stability (e.g., static balance), object control or manipulation (e.g., throwing), or locomotion involving two or more body segments (e.g., jumping) [8,9].

Stodden’s theoretical model [10] examines the relationship between PA, MC, body composition, and physical fitness. According to this model, children with low MC experience significant constraints when participating in physical activities. A lack of proficiency in FMS not only limits opportunities for physical activity engagement but also creates a barrier to becoming active [10], contributing to increased sedentary behavior. These limitations in MC and PA engagement are not merely behavioral concerns; they are directly linked to adverse health outcomes such as increased risk of obesity, poor cardiovascular fitness, and delayed neuromotor development [11,12]. In this sense, interventions that enhance MC in early childhood are not only important to promote good and balanced motor development, but also critical for promoting long-term health [13]. In this study, we refer to the participants (aged 3–4 years) as being in early childhood. Although they were recruited from a preschool setting, the concept of early childhood is used intentionally to situate them within a broader developmental framework. This period, typically defined as the period from birth to 8 years, encompasses critical phases of motor and cognitive development.

In Portugal, the Preschool Education Curricular Guidelines [14] emphasize the importance of mastering basic movements that involve locomotion and balance, e.g., climbing, running, jumping or rolling, while also promotion children’s exploratory play and self-expression. However, despite these general curricular orientations, there is no structured or mandatory physical education program in preschool, which can be a barrier to the systematic development of MC. Barriers also include limited time allocated to PA, insufficient training of early childhood educators in motor development, and contextual restrictions in school environments, such as lack of space or resources for structured movement activities [15]. Thus, there is a gap between curricular intentions and the daily reality of practice, making the integration of structured programs such as Grow+ particularly relevant.

Educators and the preschool environment play a central role in the implementation of such programs. In Portugal, private kindergartens often employ physical education teachers or exercise technicians to lead PA sessions. While in the public system, both kindergarten teachers and physical education teachers may be involved, depending on the management model of each school. The development of support materials and structured interventions enables teachers, educators and exercise coaches to implement programs consistently.

Intervention programs aimed at increasing PA levels in children have been studied by several authors, and some of these programs have combined physical training with music, i.e., dance training [16]. Dance is a comprehensive form of physical and motor activity that combines music, aerobic and rhythmic exercises, involving the exploration and practice of various skills, movement sequences, and the enhancement of coordination, spatial awareness, and memory [17]. This type of activity requires the performance of different types of movements and may involve multiple multisensory inputs (visual, auditory, tactile), which in turn can lead to various neurological adaptations, such as enhanced neural activity and neurogenesis [18]. Dance is a common physical–motor activity among preschool-aged children and is typically taught through repetitive and progressive methods to facilitate the learning and mastery of motor skills to the diverse needs of learners, including children with disabilities [19].

Although multiple dance modalities (e.g., ballet, folk or regional dances) can contribute and enhance motor development, the technical and structural demands of some of them may limit applicability at very young ages. For instance, ballet often requires postural precision, discipline, and advanced motor control that exceeds the developmental stage of preschoolers. In contrast, hip-hop is characterized by simple, rhythmic, and playful movement patterns that are easily adaptable to children aged 3–4, aligning with their coordinative capacities and need for exploratory learning. Recent studies reinforce this argument, showing that developmentally appropriate music and movement programs enhance preschoolers’ jumping and balance [20], that integrated music-and-movement interventions are particularly effective in developing both FMS and rhythmic abilities in children aged 5–6 [21], and that creative dance programs can improve proprioception, rhythm, and synchronization in preschoolers [22]. Together, these findings highlight rhythm-based activities as critical contributors to early motor competence.

Hip-hop culture is so deeply embedded in the lives of children and adolescents that the use of its elements, such as dance, can enhance physical activity (PA) levels and engagement in health interventions [23]. By blending rhythmic music with dynamic physical movement, this dance-based approach not only promotes PA but also supports the development of FMSs and broadens children’s motor repertoires [24]. It also contributes to improvements in physical fitness and body composition, such as reductions in body mass index (BMI) [25], while offering social and psychological benefits, including greater social integration and enhanced quality of life. These include improved mental health [23], better sleep quality [26], and reductions in symptoms related to somatization, obsessive–compulsive disorder, interpersonal sensitivity, depression, and anxiety [27,28]. Moreover, hip-hop practice is considered an effective strategy for increasing PA levels in children, as it is an enjoyable activity that promotes a positive, cooperative environment among young individuals [29,30]. All these hip-hop’s features make it particularly suitable choice for preschool interventions, serving simultaneously as a pedagogical strategy and a culturally relevant form of movement.

However, according to our review of the literature, despite the existence of several hip-hop-based intervention programs, only one has been specifically applied to children in early childhood. This program employed a multidimensional approach, combining PA promotion with nutritional education, its primary goal being BMI reduction. The intervention included 20 min of nutritional activity and 20 min of aerobic activity incorporating some hip-hop movements [25,31]. This scarcity contrasts the growing body of international frameworks and recommendation, such as UNESCO’s Quality Physical Education Guidelines and the World Health Organization’s Global Standards for Health-Promoting Schools, which advocate for the integration of rhythmic and expressive movement into early childhood curricula as a means of supporting holistic development. Despite the growing recognition of dance-based interventions, there remains a lack of structured programs specifically designed to explore the developmental potential of hip-hop in early childhood. Existing programs are not standardized, and the variability in their implementation makes meaningful comparisons difficult. Interventions of this type are particularly scarce among preschoolers aged 3 to 4 years. Considering the critical role of PA during this stage, and the preschool years as a sensitive period for the development of coordinative capacities and FMS [9,32], further investigation is warranted. In this context, the present study aims to analyze the effects of the ‘Grow+’ hip-hop program on children in early childhood’s motor competence and physical activity levels, as well as the perceptions of educators and caregivers regarding the children’s motor competence. Considering the previous literature, it is hypothesized that the Grow+ program significantly enhances children’s motor competence, perceived motor competence, and physical activity levels. Furthermore, it is anticipated that the program positively influences educators’ and caregivers’ perceptions of children’s motor competence.

## 2. Materials and Methods

### 2.1. Study Design

This was a non-equivalent quasi-experimental study with a self-controlled design, structured in three consecutive phases each lasting four weeks. The first phase included an initial intervention period, followed by a rest phase without intervention and a final phase with a second intervention period. To evaluate the impact of the program, four evaluation moments were conducted: one before the first intervention (O1), one after it (O2), one following the no-intervention period (O3), and one at the end of the second intervention (O4), see study design in Table 1.

The option for self-controlled design without a parallel control group, i.e., interrupted time series study design, was due to contextual constraints. To address this limitation, a non-intervention interval was introduced, which provided an internal reference for comparison, reducing some threats to validity [33].

### 2.2. Sample Characterization

A priori power analysis was conducted using G*Power (version 3.1.9.4) [34] for a repeated-measures ANOVA (within-subjects, four time points). Parameters were set as follows: effect size f = 0.25 (medium effect, as recommended by Cohen [35]), α = 0.05, desired power = 0.95, one group, four measurements, correlation among repeated measures ρ = 0.50, and nonsphericity correction ε = 1.0. The choice of a medium effect size was conservative given the absence of prior studies with the same intervention; the power level of 0.95 was adopted to minimize the probability of type II error; the correlation value of 0.50 was selected as a moderate and plausible estimate, commonly used in repeated-measures designs involving young children, especially when prior empirical data are unavailable [36]; and ε = 1.0 reflects the assumption of sphericity typically used in a priori analyses [35]. Under these assumptions, the required sample size was 36 participants. Our final sample included 37 children, meeting and slightly exceeding this requirement, and indicating that the study was adequately powered to detect effects of practical significance.

A total of 37 children participated in this study (18 boys and 19 girls), aged between 3 and 4 years (M = 4.29; SD = 0.58), all attending a public preschool in a city in central Portugal. The inclusion criterion was based on the age of the participants, specifically children aged 3 and 4 years. The exclusion criteria were: (i) having restrictions regarding physical exercise, and (ii) presenting atypical motor development. Due to illness-related absences, it was not possible to assess all children at every evaluation point. To preserve the integrity of the sample, no participants were excluded based on missing data, and as such, the sample size may vary across assessment moments.

### 2.3. Ethical Considerations

All procedures performed followed the guidelines of the 1964 Declaration of Helsinki for studies involving human participants. Informed and written consent was obtained from all legal guardians, and children’s assent was always considered. This study was approved by the Ethics Committee of Santarém Polytechnic University, under approval number 5A-2023 ESDRM.

### 2.4. Intervention Program

The intervention consisted of applying the Grow+ program, a hip hop program previously developed and validated for pre-school children [37]. This program included three sessions per week, each lasting 30 min. The 30-minute session duration was chosen according to the target age group and the motor development stage of the participants, who were between 3 and 4 years old and in the fundamental motor phase [32]. Each session followed a structured progression, beginning with basic locomotor and stability skills (e.g., walking, running, jumping) and gradually integrating more complex sequences that required rhythm, balance and coordination. The program content was aligned with the Portuguese Preschool Education Curricular Guidelines [14] and motor development frameworks [9,38], ensuring developmental appropriateness for children aged 3–4 years. The program content and activities were specifically designed to be culturally appropriate and tailored to the cognitive and motor abilities of this age group. Attendance was recorded for all participants throughout the intervention.

The hip-hop sessions were led by two certified exercise professionals, both enrolled in a Master’s program in Physical Activity and Health and possessing three years of experience teaching hip-hop to children. Instructors received specific training supported by a standardized manual and instructional videos. This training included structured guidance on choreography progression, motor skill targets, and pedagogical strategies, ensuring that all sessions followed a consistent format. Instructors were also provided with detailed session plans and visual materials to support uniform delivery. Although no formal fidelity checklist was applied in all sessions, intervention consistency was supported by standardized instructor training, detailed session plans, and the use of a validated program [37].

The program focused on simple and short choreographies. Three of the four choreographies were taught and repeated during the first intervention phase, and a slightly more complex choreography was introduced in the second phase. The motor skills (i.e., choreographic steps) included in the program and choreographies were selected based on the following criteria: (i) steps of low to moderate difficulty, based on foundational motor skills such as walking, jumping, and squatting, with a level of difficulty adjusted to the target population; this methodological choice allowed for the exploration of basic skills essential for the acquisition of more specialized skills in the future. (in hip hop and other forms) [9]; (ii) inclusion of stylistic and identity-based variations characteristic of hip hop; this inclusion grounded the chosen dance style and provided children with new motor experiences [23,28]; (iii) exploration of different movement planes and execution levels (i.e., low, medium, and high levels); this approach challenged the children’s coordination, helping them improve their ability to perform complex and synchronized movements. Variation in movement planes and execution levels also added diversity to the sessions, making them more engaging and stimulating.

### 2.5. Procedures and Protocols

All assessments were conducted by three trained and experienced professionals specialized in early childhood motor development, under the same environmental conditions, in the same location, using the same materials and procedures. The following section outlines the protocols used to assess the several variables under analysis.

(i)Perceived physical activity level

The perceived physical activity level was assessed using the Pictorial Questionnaire of Children’s Physical Activity [3]. This questionnaire consists of six closed-ended questions and a pictorial response scale. The first five questions are designed to determine the child’s perceived physical activity level over the past seven days. These include: first question- during the week (Monday to Friday); second question- on weekends; third question- during school breaks; fourth question- when not at school, and during physical activity or motor skills classes [39]. All response options are presented as images, each illustration representing a specific level of physical activity. The illustrations are gender-neutral and do not include facial expressions. The first image represents a “sedentary” state (1 point); the second, a “lightly active” state (2 points); the third, an “active” state (3 points); and the fourth, a “vigorous” state (4 points). The sixth question focuses on the mode of transportation used to get to school, with the following response options: walk, bus, car, bicycle, or other [3,39].

The final perceived physical activity level score is obtained by calculating the arithmetic mean of the scores from the first five questions. The closer the value is to four, the more physically active the child is perceived to be; conversely, the closer the value is to one, the more the child tends to perceive themselves as sedentary [39].

(ii)Motor Competence

Motor competence was assessed using the MCA battery [7] (Luz Et Al., 2016), a quick-to-administer and validated test battery for the Portuguese population. This battery includes three domains, i.e., locomotion, stability, and manipulation, each of which comprises two tests. In the locomotor domain, children performed the Shuttle Run (SHR), involving four 10-m sprints while transporting and switching wooden blocks, and the Standing Long Jump (SLJ), where they jump forward with both feet simultaneously to measure distance. The stability domain included the Platform Transfer, in which children move laterally between two small platforms for 20 s, and the Lateral Jump, where they jump side-to-side over a beam for 15 s, with only correct jumps being counted. In the manipulative domain, children completed the Throwing Velocity test by throwing a tennis ball against a wall at maximum speed, and the Kicking Velocity test, kicking a soccer ball with maximum force; in both cases, the highest velocity from three trials was recorded using a radar gun Pro II Stalker Radar Gun.

Each test was scored using age- and sex-specific percentiles, and the overall motor competence score was calculated as the average of the three domain scores [7,40].

(iii)Perceived Motor Coordination

To assess children’s perceived motor coordination, the Questionnaire on Motor Coordination Development in Children Aged 3 to 5 Years for Parents and Educators was administered [41,42]. This instrument relies on parent or educator reports comparing the motor performance of the child to that of peers of the same age, using a 5-point Likert scale. The questionnaire provides a standardized method for measuring a child’s coordination development in everyday and functional activities. It consists of 15 items grouped into three distinct factors: the first factor, “Control During Movement,” includes items related to motor control while the child or an object is in motion; the second factor, “Fine Motor and Handwriting,” involves items addressing fine motor skills and early writing abilities; and the third factor refers to “General Coordination.” In this study, the same version of the questionnaire was completed independently by both the children’s legal guardians and their preschool educators, allowing for a comparative perspective on perceived motor coordination across home and school contexts. In addition to evaluating coordination development, this tool allows for the screening of Developmental Coordination Disorder (DCD). Based on a total score of 75 points, participants are classified into the following categories: Class 1—not DCD (51 or more points and no motor delay); Class 2—at risk for DCD (between 41 and 50 points with slight motor delay); and Class 3—probable DCD (40 points or less with significant motor delay) [41,42].

(iv)Body Composition

To assess body composition, measurements of body mass and height were performed. Body mass was measured in the morning using a calibrated placed on a firm, level surface (Tristar WG-2424, Tristar Europe B.V., Tilburg, The Netherlands), with a precision of 0.1 kg). Height was measured with a portable stadiometer (Seca 213, Seca GmbH & Co. KG, Hamburg, Germany), with a precision of 0.1 cm, while participants maintained the anthropometric position and performing a deep inhalation. Both body mass and height measurements followed the protocols of the International Society for the Advancement of Kinanthropometry (ISAK) [43]. Finally, Body Mass Index (BMI) was calculated and classified according to the World Health Organization (WHO) criteria [44].

### 2.6. Data Treatment and Variables

Data treatment was performed using IBM SPSS Statistics (version 28). Descriptive statistics (mean, standard deviation, median, minimum, and maximum) were calculated for all variables and for sample’s characterization. In addition, 95% confidence intervals were estimated for the relevant variables. To examine changes across the four evaluation moments, repeated measures ANOVA were used. The assumption of sphericity was verified using Mauchly’s test, and Bonferroni adjusted pairwise comparisons were conducted when significant effects were found. In addition, Spearman correlation coefficients were calculated to explore associations between variables such as age, motor competence, and perception scores. The significance level was set at *p* ≤ 0.05.

In this study, children’s age was considered a fixed variable, while the preschool context functioned as a control variable to ensure consistency across the sample. The independent variable was participation in the hip-hop intervention program. Four dependent variables were analyzed in line with the study’s aims: motor competence, perceived motor competence, perceived motor coordination, and perceived physical activity levels.

## 3. Results

### 3.1. Motor Competence

The descriptive statistics for the motor competence variable, including the mean (M), standard deviation (SD), median (Md), minimum (Min), and maximum (Max) at the four evaluation moments, are presented in the table below (Table 2).

A progressive increase in motor competence (MC) percentiles was observed across the four evaluation moments, as shown in the boxplot (Figure 1). This trend is reflected in the gradual rise in the median values. The minimum and maximum scores also show relative stability, suggesting a consistent performance range within the group.

Considering that decimal age could serve as a general indicator of motor competence, under the assumption that increased age corresponds to greater opportunities for motor experience, a Spearman correlation analysis was conducted to examine the association between these two variables (Table 3).

At the first observation, performed prior to the intervention, a statistically significant association occurred between decimal age and motor competence, with older children presenting higher motor competence scores. However, this association was no longer statistically significant in the second observation, conducted after the first intervention period, and the correlation further diminished in subsequent assessments.

Motor competence showed a significant change over the four observation moments (*F*(3, 84) = 5.744, *p* = 0.001, *η*^2^ = 0.170) (Table 4), indicating a meaningful evolution throughout the intervention period. A significant increase in motor competence was observed between the first (O1) and second (O2) evaluation moments, corresponding to the pre- and post-assessment of the first intervention period (cf. Table 1), suggesting improvement in motor skills promoted by participation in the program. No significant differences were found between the second (O2) and third (O3) evaluations, which followed a period without intervention. Although the mean values indicate a slight increase in motor competence during this interval, the change was not statistically significant, suggesting that performance remained relatively stable in the absence of the program. Following the second intervention period, no significant improvements were observed (O3 Vs. O4). However, when the second post-assessment (O4) was compared with pre-assessment (O1) and the second post-assessment (O2), significant improvements were evident. This suggests that the second intervention phase may require revision and enhancement to provide more effective stimuli for promoting motor skill development.

### 3.2. Adult Perception of Children’s Motor Coordination

Motor coordination perception (PMCoor) was assessed by the children’s guardians and preschool educators. The descriptive statistics of this variable, expressed by the total score of the specific questionnaire applied [41,42], are presented in Table 5.

It can be observed that there was an average increase in scores between observation moments 1 and 4 (and between moments 2 and 4 for educators), accompanied by a homogenization of the scores (i.e., a decrease in standard deviation).

Given the interrelationship between decimal age and motor competence—and their combined influence on the perception of motor coordination [10], Spearman correlation analyses were conducted to explore three key associations: (i) between decimal age and adult-reported motor coordination scores (Table 6), (ii) between the coordination perception scores provided by guardians and educators across different observation moments (Table 7), and (iii) between the directly assessed motor competence percentiles (via the MCA battery) and the educators’ indirect evaluations of motor coordination (Table 8).

The statistical results reveal no relationship between decimal age and caregivers’ perception of their children’s motor coordination in any observation moment. This indicates that, from the caregivers’ perspective, the child’s chronological age does not influence how they evaluate their motor coordination. However, when analyzing the evaluation made by early childhood educators, the opposite occurs (Table 6); educators assess motor coordination development according to the child’s decimal age, consistent with the children’s own perception at the first observation moment, where a direct association between decimal age and perceived motor competence was found. Correlations between caregivers’ and educators’ responses across all moments, showed that significant association only occurred in the last observation moment.

### 3.3. Physical Activity

The potential association between decimal age and perceived physical activity practice was examined (Table 9). A significant negative association was observed only at the third observation moment, indicating that older children engaged in a lower volume of physical activity compared to younger children, although this association was weak and not sustained at the fourth moment.

The questionnaire used to assess the children’s perception of their physical activity levels included four questions. The children showed consistency in their answers, both throughout the different assessment moments for each question and between different questions at the same moment (see Table 10). At the second observation point, which took place after the first period of intervention with hip hop, a significant association was identified between Question 1 (“During the last week, from Monday to Friday, I was mostly…”) and Question 4 (“During the last week, when I wasn’t at school, I was mostly…”). There was no such association at the first assessment point. In addition, associations were observed between: Question 1 at moments two and four; Question 1 at moments two and three; Question 1 and Question 5 (“During physical activity or motor skills classes…”); and between Question 2 (“During the weekend…”) and Question 5. These patterns are detailed in Table 10.

## 4. Discussion

This study aimed to analyze the effects of the “Grow+” hip-hop intervention program on four key variables in children in early childhood: motor competence (MC), perceived motor coordination (PMCoor), and perceived physical activity (PA) levels. The intervention was designed to explore whether a structured, rhythmic movement program rooted in hip-hop culture could support both motor and psychosocial development during a sensitive period of early childhood.

The choice of hip-hop as the movement framework for the Grow+ program was deliberate and theoretically grounded, though other dance modalities such as ballet or regional dances could also improve motor competence and promote several benefits. These styles often rely on more formal techniques, longer sequences, or culturally specific codifications that may reduce inclusivity and immediate participation among very young children. Hip-hop, by contrast, is rhythmically driven, and socially expressive, with movement motifs that can be simplified or scaled to each child’s developmental stage. This adaptability is consistent with recent evidence that rhythm-based programs enhance FMS and balance in preschoolers [24,45]. Moreover, the playful and culturally familiar context of hip-hop increases motivation and engagement, which are critical drivers of adherence to movement interventions at this age [3,23].

The Grow+ program aligns with the Stodden’s theoretical and conceptual framework [10], which argues that motor competence influences physical activity through direct and mediated pathways, including perceived motor competence and physical fitness. According to this model, rhythmic and multisensory activities, such as dance and consequently hip-hop, can be catalysts for motor development, improving coordination, timing, and body awareness. Hip-hop, with its emphasis on beat synchronization, expressive movement and repetition, can activate these mechanisms, supporting both motor competence and its perception. Multisensory stimulation, a central component of dance-based interventions, simultaneously involves visual, auditory and kinesthetic systems, promoting neural integration and motor learning [24]. The expressive movement typical of hip-hop promotes emotional engagement and self-expression, which, although not evaluated in this study, is known to increase motivation, a key mediator in Stodden’s model [10]. These elements may contribute to the improvements observed.

Regarding MC, there was a significant difference across the observation moments, associated with different decimal age (*F*(3, 84) = 5.744, *p* = 0.001, *η*^2^ = 0.170). Thus, these findings suggest that participation in the intervention may have contributed to the changes in MC levels among the children, although other factors such as natural maturation or school context may also have played a role, substantially reducing the effect previously linked to greater motor experience, as indicated by higher decimal age. If this hypothesis is correct, younger children, in terms of decimal age, attained similar MC levels to older children, implying that at these ages, motor practice opportunities may be more important than age. The results also suggest that the program may have had a similar impact across this age range, but this interpretation should be made cautiously given the absence of a control group, practically independent of each child’s chronological age. The observed reduction in the correlation between decimal age and motor competence following the intervention is consistent with the theoretical framework proposed by Stodden et al. [10], which argues that structured motor practice can mitigate age-related disparities in motor competence during early childhood. In the present study, younger children reached similar motor competence levels to their older peers after participating in the Grow+ program, suggesting that the intervention may have equalized motor development opportunities across age groups. While we acknowledge that measurement variability or confounding factors cannot be entirely excluded, the consistency of the results across multiple assessment points and their alignment with established theoretical models support the interpretation of a meaningful intervention effect rather than a spurious association.

The interruption interval tests not only the children’s ability to retain what they have learned but may also serve as a pause during which the neuromotor system reorganizes [10]. This break between the two intervention periods allowed, from an experimental design perspective, to assess both the retention efficiency and the delayed effect of the hip hop intervention program. These data suggest that the implemented program may have contributed not only to the preservation of the attained competence levels, but also to reinforcing them, indicating that effective learning persisted despite the potential for regression. Also, the comparison between the second and fourth observation moments reveals a significant difference, i.e., after the second intervention period there was a further improvement in MC levels. This may suggest that the intervention program, as implemented, had the potential to boost the average motor competence level of the children in this sample. However, since there is no significant difference between the last two observation points, this may indicate that the children reached a plateau in their MC level, meaning the program’s ability to further enhance MC may have been exhausted. This result points to the need for further increments in the “Grow+” program to test whether it can continue to add value to the motor competence of children in this age group.

Given that children’s motor competence remained stable in the absence of hip hop intervention and improved significantly when the intervention was implemented, it is plausible that both the duration of practice and the retention periods were appropriately tailored to their developmental needs. This temporal regulation of practice and rest may represent an additional factor to consider in the implementation of the program. The present results are consistent with the hypothesis that the hip hop “Grow+” program could contribute to improvements in MC in preschool-aged children, although causal claims should be qualified. It is also possible that the absence of significant differences between O3 and O4 reflects other factors, such as a ceiling effect in motor competence, reduced sensitivity of the assessment tools at higher performance levels, or natural developmental variability. These findings are aligned with Stodden’s model, which predicts that during early childhood (ages 2 to 5), physical activity practice (of moderate to vigorous intensity) directly influences motor competence [10,46]. However, the Grow+ program was based on rhythmic motor activities performed at a pace that did not necessarily require vigorous physical effort. Despite this, the results suggest that Grow+ was effective in promoting improvements in MC even in the absence of high intensity. This raises the hypothesis that structured and engaging rhythmic movements, such as hip-hop dance, can stimulate motor development through other mechanisms, such as repetition, coordination and multisensory involvement. Aligned with this hypothesis, the pictorial questionnaire data revealed a progressive increase in children’s perceived physical activity levels throughout the intervention. After the first intervention period, new associations emerged between different practice contexts (e.g., weekdays vs. weekends, school vs. home), suggesting that the program may have influenced not only motor competence, but also children’s perception and awareness of when and how physically active they are. Although the instrument used assesses perception and not the objective practice of PA, which represents a limitation, the consistency and evolution of the responses over time reinforce the idea that the intervention had a wider impact on children’s involvement with movement. Thus, the perception of physical activity, especially in this age group, can be a relevant indicator of real involvement and motivation to practice, and is therefore relevant to motor development.

It was explored how adults, both caregivers and educators, perceived the children’s motor coordination throughout the intervention. Similarly to Cordovil and Barreiros’ [47] previous study, caregivers tend to have less accurate perception of what children are physically capable of compared to early childhood educators. Another study supports this understanding by showing that parents often overestimate their children’s abilities, revealing a limited capacity to objectively evaluate their behavior [48].

Additionally, there are correlations between caregivers’ questionnaire scores across different observation moments (Table 7), suggesting that caregivers consistently believe they are responding accurately, maintaining the same response pattern even when actual improvements occurred. Parents who gave lower scores at the first observation also tended to do so in subsequent assessments despite real progress. This pattern also appeared in the correlations between educators’ questionnaire scores. The significant association found between educators’ evaluations and children’s motor competence at the first observation, indicates a better evaluative ability compared to caregivers. However, this association disappears in the following observations. These results suggest that although children continued to improve their motor competence, educators maintained a consistent perception of the children’s motor coordination over time, resulting in a loss of significance in subsequent correlations between their evaluations and the children’s motor competence. This may reflect an anchoring effect, where educators’ initial impressions of the children’s motor abilities served as a cognitive reference point, influencing subsequent evaluations regardless of actual progress [49]. Such anchoring may have been reinforced by their lack of effective involvement in the program, which prevented them from better appreciating children’s improvements, or these improvements were more evident in the specific hip-hop motor abilities than in everyday preschool motor activity. To mitigate this, future implementations of the Grow+ program should consider involving educators more actively, e.g., through co-facilitation, structured observation, or training in motor development literacy. These educators’ active participation and integration can also contribute to catalyzing the program’s results.

In line with the study’s aim of assessing the impact of the Grow+ program on preschoolers’ perceived physical activity levels, the results suggest that the intervention may have influenced how children engaged with movement across various daily contexts. The emergence of associations between the perception of PA during the week and outside of school after the first intervention period suggests that practicing hip-hop may have influenced the children’s physical activity habits beyond the structured sessions. The absence of these associations at the first moment, followed by their appearance at subsequent moments, may indicate a tendency for levels of physical activity to increase during the program. In addition, the evolution of associations between physical activity at the weekend and during motor classes suggests that the intervention may have contributed to a more integrated perception of physical activity in different contexts of daily life. This pattern may reflect a broader impact of the program on the way children engage with movement, and their awareness of when they are physically more active. The findings align with Romero’s [50] research, which concluded that hip-hop dance implementation emerged as an effective strategy for increasing physical activity, among Mexican-American adolescents.

From the point of view of health promotion, the results obtained suggest that the Grow+ program can be used as a tool to promote health from pre-school age onwards, although further controlled studies are needed to confirm its effectiveness and the results’ generalization. This stage of development, i.e., second childhood, is the critical period for the development of various coordinative abilities and the acquisition of foundational motor skills [10,51]. This in turn makes it an extremely important phase for consolidating health-related behaviors that can last throughout life [10]. By involving children in structured, enjoyable motor activities adjusted to their level of development, such as hip-hop in the Grow+ program, we can promote not only motor skills but also reduce sedentary behavior [52]. It is worth emphasizing that a sedentary lifestyle is one of the main risk factors associated with childhood obesity, low cardiorespiratory fitness and delayed motor development [53].

Considering that the program, in addition to promoting CM and PA, also promoted improvements in the perception of physical activity and motor competence, it is possible to assume that the children improved their body awareness and physical self-efficacy, aspects that are fundamental to psychological well-being and sustained adherence to active lifestyles [54,55]. In this way, the Grow+ program seems to support a holistic model of child health.

Although the main objective of this study was not to integrate the Grow+ program into the curriculum or to evaluate this possibility, the results suggest that structured and rhythmic interventions may be aligned with the main developmental goals in early childhood education. The improvements observed in motor competence and perceived levels of physical activity are consistent with the competencies promoted in preschool curricula, such as gross motor development, body awareness, and socio-emotional engagement through movement [56,57]. Although further research is needed to assess the feasibility of implementing such programs in daily preschool routines, the success story of Grow+, its simplicity, adaptability, and short session duration give a positive indication of its viability and replicability.

This study has several limitations that should be acknowledged. First, the quasi-experimental self-controlled design without a parallel control group limits causal inferences, although the inclusion of a non-intervention interval provided an internal reference to mitigate some threats to validity [33]. Second, although a priori power analysis indicated that the sample size was sufficient to detect medium effects, the relatively small and context-specific sample restricts the generalizability of the findings. Third, while the intervention was based on a validated and standardized program, no formal fidelity checklist was systematically applied; instead, fidelity was supported indirectly through instructor training, session plans, and adherence to a validated manual. Fourth, the assessors were not blind to group allocation, which may have introduced observer bias, although children themselves were blind to the study hypotheses and standardized protocols were used to minimize this risk. At last, PA outcomes were based on validated self-perception measures rather than objective monitoring, which can limit conclusions. Future studies should address these limitations by incorporating blinded assessments, objective PA measures, larger and more diverse samples, and systematic fidelity monitoring within controlled designs to confirm and extend the present findings.

## 5. Conclusions

The Grow+ program showed that movement, when made fun and meaningful through hip-hop, can be a powerful tool for supporting children’s physical and motor development. The present results showed that the Grow+ hip-hop program significantly improved motor competence, perceived motor competence, and perception of physical activity levels in children in early childhood. These improvements occurred during the intervention phases and were not solely explained by age, suggesting the program’s effectiveness in promoting motor development, independent of chronological age, reinforcing the concept that a motor age is a more accurate measure of motor development than chronological age. Nevertheless, given the quasi-experimental design and absence of a control group, these findings should be interpreted with caution.

The hip hop intervention also affected the children’s perception of their own motor competence. Although initially, chronological age was associated with this perception, this correlation gradually diminished and disappeared by the final assessment, with younger children perceiving their motor competence independently of age, and at a similar level to that of older children.

Regarding adults’ perception of the children’s motor coordination, a discrepancy was observed between parents and educators. Parents showed a misaligned perception, while educators’ perceptions were influenced by the children’s chronological age. Both groups maintained consistent perceptions throughout the assessments, failing to recognize the improvements reported by the children. These findings may reflect limited awareness or training in motor development among both groups. In both cases, raising awareness about this dimension of child development appears necessary.

Overall, the Grow+ program appears to be a promising and engaging strategy not only for stimulating motor development and adherence to physical activity in childhood, but also as a tool for promoting health. By integrating rhythmic movement in a playful and structured way in the context of pre-school education, the program contributes to the creation of healthy habits from an early age, helping to combat sedentary lifestyles and foster children’s physical, emotional, and social well-being. This approach reinforces the importance of including teaching proposals that value the body in movement as an integral part of health-promoting education. Future studies with controlled designs are recommended to confirm these findings and further explore the program’s educational integration and long-term impact.

## Figures and Tables

**Figure 1 healthcare-13-02518-f001:**
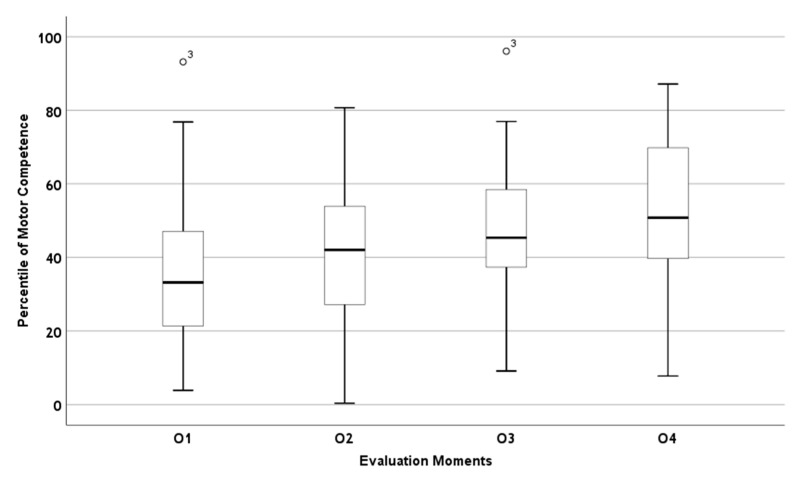
Boxplot graph of the percentile of motor competence throughout the evaluation moments (O1—observation 1, O2—observation 2, O3—observation 3, O4—observation 4).

**Table 1 healthcare-13-02518-t001:** Study design of the “Grow+” program.

O1	X	O2	—	O3	X	O4

Notes: O—evaluation moment, X—intervention period,——rest phase without intervention.

**Table 2 healthcare-13-02518-t002:** Descriptive statistics of the motor competence percentile over the four evaluation moments.

Motor Competence Percentile	M	SD	Md	Min	Max
Observation 1	35.90	19.80	33.18	3.87	93.17
Observation 2	41.41	18.49	42.00	0.39	80.68
Observation 3	46.01	18.59	45.32	9.13	96.08
Observation 4	50.87	18.70	50.76	7.76	87.14

**Table 3 healthcare-13-02518-t003:** Spearman correlation coefficients (*r_s_*) between decimal age and motor competence percentile, across the four evaluation moments, for the entire sample.

Observation Moments	*r_s_*	*p*	95% CI	
			Lower	Upper
Observation 1	0.404	0.020	0.059	0.662
Observation 2	0.329	0.066	−0.033	0.614
Observation 3	−0.040	0.819	−0.377	0.306
Observation 4	0.028	0.872	−0.313	0.362

Notes: *r_s_*—Spearman correlation, *p*—significance, CI—confidence interval.

**Table 4 healthcare-13-02518-t004:** Pairwise comparisons of motor competence percentiles between evaluation moments (O1 to O4).

Motor Competence		Test Statistic	*p*	95% CI	
				Lower Limit	Upper Limit
Observation 1	O2	1.929	0.048 *	−10.980	−0.029
	O3	2.489	0.002 *	−17.180	−3.045
	O4	2.548	0.000 ***	−22.198	−7.733
Observation 2	O3	1.794	0.095	−9.701	0.485
	O4	2.360	0.002 **	−16.160	−2.762
Observation 3	O4	2.294	0.260	−11.366	1.659

* ≤0.05; ** ≤ 0.01; *** ≤ 0.001.

**Table 5 healthcare-13-02518-t005:** Descriptive analysis of perceived motor coordination ratings, as assessed by guardians and educators, across the four evaluation moments.

Observation/Evaluators	M	SD	Md	Min	Max
Observation 1/Guardians	63.33	6.77	64.50	50.00	75.00
Observation 2/Guardians	66.89	6.15	69.00	53.00	75.00
Observation 3/Guardians	66.89	6.15	69.00	53.00	75.00
Observation 4/Guardians	67.83	5.32	68.50	55.00	75.00
Observation 2/Educators	56.22	11.86	62.00	36.00	72.00
Observation 3/Educators	56.22	11.87	62.00	36.00	72.00
Observation 4/Educators	59.50	10.91	64.50	39.00	72.00

**Table 6 healthcare-13-02518-t006:** Spearman correlation coefficients (*r_s_*) between decimal age and adult perception of motor coordination score, per evaluation moment, for the entire sample.

Observation/Evaluators	*r_s_*	*p*	95% CI	
			Lower	Upper
Observation 1/Guardians	−0.024	0.899	−0.384	0.343
Observation 2/Guardians	0.040	0.874	−0.447	0.509
Observation 3/Guardians	0.117	0.577	−0.303	0.499
Observation 4/Guardians	0.036	0.839	−0.315	0.379
Observation 2/Educators	0.484	0.003	0.170	0.709
Observation 3/Educators	0.482	0.003	0.173	0.705
Observation 4/Educators	0.428	0.009	0.106	0.669

Notes: *r_s_*—Spearman correlation, *p*—significance, CI—confidence interval.

**Table 7 healthcare-13-02518-t007:** Spearman correlation coefficients (*r_s_*) between the scores for the perception of motor coordination reported by guardians and educators across different observation moments.

Observations/Evaluators	*r_s_*	*p*	95% CI	
			Lower	Upper
O1 vs. O2/Guardians	0.681	0.002	0.300	0.874
O1 vs. O3/Guardians	0.637	0.001	0.282	0.838
O1 vs. O4/Guardians	0.729	0.000	0.480	0.869
O2 vs. O3/Guardians	1.000	0.000		
O2 vs. O4/Guardians	0.763	0.000	0.448	0.909
O3 vs. O4/Guardians	0.761	0.000	0.507	0.894
O2 vs. O3/Educators	1.000	0.000		
O2 vs. O4/Educators	0.775	0.000	0.588	0.883
O3 vs. O4/Educators	0.787	0.000	0.612	0.888

Notes: *r_s_*—Spearman correlation, *p*—significance, CI—confidence interval.

**Table 8 healthcare-13-02518-t008:** Spearman correlation coefficients (*r_s_*) between motor coordination perception scores reported by guardians and educators across different observation moments.

Observation Moments	*r_s_*	*p*	95% CI	
			Lower	Upper
Observation 2	0.630	0.000	0.351	0.806
Observation 3	0.272	0.114	−0.078	0.562
Observation 4	0.213	0.212	−0.134	0.514

Notes: *r_s_*—Spearman correlation, *p*—significance, CI—confidence interval.

**Table 9 healthcare-13-02518-t009:** Spearman correlation (*r_s_*), for 95% confidence interval (CI), between the decimal age of the different observation moments and the weekly volume of physical activity practice.

Decimal Age	*r_s_*	*p*	95% CI	
			Lower	Upper
Observation 1	−0.272	0.103	−0.555	0.067
Observation 2	−0.275	0.115	−0.568	0.080
Observation 3	−0.339	0.043	−0.607	−0.001
Observation 4	−0.252	0.138	−0.543	0.093

Notes: *r_s_*—Spearman correlation, *p*—significance, CI—confidence interval.

**Table 10 healthcare-13-02518-t010:** Spearman correlation (*r_s_*), for 95% confidence interval (CI), between responses to the “Child Physical Activity Pictorial Questionnaire”.

Questions_Observation Moments	*r_s_*	*p*	95% CI	
			Lower	Upper
Q1_O1\Q1_O4	0.493	0.004	0.170	0.720
Q4_O1\Q3_O2	0.371	0.037	0.014	0.643
Q1_O2\Q4_O2	0.568	0.000	0.274	0.764
Q1_O2\Q5_O2	0.457	0.007	0.130	0.694
Q1_O2\Q1_O3	0.398	0.020	0.059	0.655
Q1_O2\Q1_O4	0.391	0.022	0.050	0.650
Q1_O2\Q2_O4	0.475	0.005	0.153	0.706
Q2_O2\Q5_O2	0.432	0.011	0.100	0.678
Q4_O2\Q2_O4	0.429	0.011	0.096	0.676
Q1_O3\Q1_O4	0.395	0.017	0.067	0.647
Q2_O3\Q3_O4	0.526	0.001	0.229	0.733
Q3_O3\Q4_O3	0.516	0.001	0.217	0.727
Q4_O4\Q5_O4	0.388	0.019	0.058	0.641

Notes: *r_s_* –Spearman correlation, *p*—significance, CI—confidence interval, O1—observation 1, O2—observation 2, O3—observation 3, O4—observation 4, Q1—question 1 “During the last week, from Monday to Friday, I was mostly…”, Q2—question 2 “During the weekend, I was mostly…”, Q3—question 3 “Last week, during breaks at school, I was mostly…”, Q4—question 4”During the last week, when I wasn’t at school, I was mostly…”.

## Data Availability

Data availability is possible upon request and with the approval of the institutional ethics committee.

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
