# Peer review of "A Quasi-Experimental Hip-Hop-Based Program to Improve Motor Competence and Physical Activity in Preschoolers in Portugal: The “Grow+” Program"

_healthcare, 2025, doi:10.3390/healthcare13192518_

Round 1

Reviewer 1 Report

Comments and Suggestions for Authors

The following comments are intended to provide constructive feedback to enhance the scientific rigor, conceptual clarity, and educational relevance of the manuscript. The observations address key aspects of the study—including its theoretical framing, methodological design, interpretation of results, and formal presentation—with the aim of supporting the authors in refining their work for publication. Each point is offered with the intention of strengthening the manuscript’s contribution to the field of early childhood motor development and educational intervention research.

General Observations

The sample size (n = 37) is relatively small, which limits the generalizability of the findings. While this limitation can be addressed in the main body of the article, it would be advisable to briefly acknowledge it in the abstract. Additionally, the study does not clarify whether the order of the intervention phases was randomized or whether any external variables were controlled during the rest period.

Furthermore, the decision to assess motor competence (MC), perceived motor competence (PMC), perceived motor coordination (PMCoor), and physical activity (PA) in the context of a hip-hop intervention for early childhood education may appear self-evident. This raises the question of whether the intervention functions more as a pedagogical innovation or as a tool with broader educational implications. Would similar outcomes have emerged from programs based on regional dances, ballet, or other movement forms?

In this regard, while the use of dance is justified, the specific choice of hip-hop is not sufficiently supported—particularly in the introduction and discussion. The rationale for selecting hip-hop over other dance modalities should be clearly articulated, especially considering the developmental stage of the participants.

The manuscript uses the terms early childhood and preschool interchangeably. Greater terminological precision is recommended. The lack of clear distinction between these terms may lead to misinterpretations regarding the scope and applicability of the findings. It is suggested that the authors define the chosen term explicitly and justify its use in relation to the study design, ensuring conceptual coherence and population specificity. In particular, the use of early childhood is encouraged, as it frames the intervention within a broader developmental perspective. This choice supports a holistic understanding of motor learning and aligns with international educational frameworks focused on comprehensive early development.

Title

The title could benefit from including a reference to the study design or type of intervention, and the name of the program might be more appropriately placed in the subtitle. Additionally, it would be helpful to specify the geographical or educational context of the intervention.

Abstract

The abstract would benefit from improved conceptual clarity and terminological precision:

  • Both perceived motor competence (PMC) and perceived motor coordination (PMCoor) are mentioned, but the distinction between these constructs is not clearly explained. A brief conceptual clarification or justification for evaluating them separately would be helpful.
  • The term self-controlled may be misleading. If the study follows a pre-post design without a control group, it would be clearer to describe it as a within-subjects quasi-experimental design.
  • The phrase positive trend is vague. Does it refer to a non-significant improvement or a linear progression? Greater specificity is recommended.

Keywords

Keywords should align with established thesauri and avoid redundancy with terms already present in the title or abstract. More concise and specific alternatives are suggested:

  • hip-hop dance, early childhood, motor skills, physical activity, intervention study

Introduction

The use of Stodden’s theoretical model provides a solid and relevant framework for analyzing the constructs under study. Similarly, the justification for using dance as a pedagogical and motor tool is appropriate, including its neurological and psychomotor benefits.

However, the focus on hip-hop requires further elaboration. While general benefits of hip-hop are mentioned, the rationale for selecting this specific modality over others (e.g., ballet, traditional dances, creative movement) for 3–4-year-old children is not sufficiently developed. Key questions remain:

  • What characteristics make hip-hop particularly suitable or effective at this developmental stage?
  • Has the cultural, cognitive, or motor appropriateness of hip-hop for this age group been considered?
  • Could the use of hip-hop be more closely linked to pedagogical innovation than to a solid educational rationale?

Moreover, the introduction does not address how this intervention aligns with the early childhood education curriculum or what specific educational needs it aims to meet. This is essential for contextualizing the intervention and assessing its transferability to other socio-educational settings:

  • What current barriers exist to motor competence development in early childhood education?
  • What role do educators and the school environment play in implementing such programs?

The transition from general dance to hip-hop is somewhat abrupt. A smoother transition is recommended to explain how the discussion moves from a broad category (dance) to a specific form (hip-hop). Additionally, the literature review on hip-hop is extensive but lacks critical depth. While many benefits are cited, potential limitations—such as lack of standardization, variability in implementation, or limited evidence in preschool populations—are not addressed.

A simple search using “early childhood” AND “hip-hop” in Web of Science reveals more research than is suggested in the introduction. It is therefore recommended to revise the literature to ensure that a significant portion of the references focus specifically on the 3–5 age range. Studies that critically examine the use of hip-hop in educational contexts—not just as a health or entertainment tool—should be included. Furthermore, references to international normative and pedagogical frameworks supporting the inclusion of rhythmic movement in early childhood education would strengthen the argument.

Methods

This is the most robust section of the manuscript. It is clearly structured and provides detailed descriptions of the study design, instruments, and evaluation conditions. The use of repeated measures ANOVA and Spearman correlations is appropriate.

However, the section could benefit from improved narrative economy. The level of detail, while thorough, may be excessive for a specialized reader. For instance, the item-by-item description of questionnaires and motor tests could be summarized or moved to an appendix. The essential information for replicability should be retained, but redundancies should be avoided.

The description of the design as self-controlled is not accompanied by a justification for the absence of a control group. It would be helpful to explain whether logistical, ethical, or contextual constraints prevented its inclusion. Additionally, the potential influence of natural maturation effects in 3–4-year-old children should be briefly discussed.

Finally, the study evaluates multiple dimensions (MC, PMC, PMCoor, PA, BMI), which enriches the analysis but may dilute the focus. Are all these variables directly aligned with the objectives of the hip-hop program? For example, is the inclusion of BMI justified given the short duration of the intervention? If so, a brief rationale would be appropriate.

Discussion

The discussion offers a detailed interpretation of the results, with attempts to connect them to theoretical and pedagogical frameworks. The effort to analyze the data from multiple dimensions—motor, perceptual, social, and educational—is commendable, as is the intention to project the program’s impact beyond the classroom.

However, several aspects warrant further refinement:

  1. Causal attributions should be moderated.
    The discussion tends to attribute observed changes directly to the intervention without sufficiently considering alternative explanations such as natural maturation, school context, or the Hawthorne effect. Given the absence of a control group, causal claims should be carefully qualified.
  2. Theoretical integration could be deepened.
    Although Stodden’s model is referenced, the discussion does not elaborate on how hip-hop specifically fits within this framework. It would be valuable to explore mechanisms such as multisensory stimulation, rhythm, and expressive movement. Including references on dance in early childhood education and the role of rhythmic movement in psychomotor development would strengthen the theoretical grounding.
  3. Curricular alignment is underdeveloped.
    While the program is said to promote health and well-being, the discussion does not address how it aligns with early childhood education curricula or what specific educational competencies it supports. A reflection on the feasibility of implementing such programs in real-world school settings is recommended.
  4. Adult perceptions are underexplored.
    The divergence between educators’ and caregivers’ perceptions is intriguing but not sufficiently unpacked. What does it mean that educators did not perceive improvements? How might teachers be more actively involved in the intervention? These questions have important implications for the educational value and sustainability of the program.

In summary, the discussion would benefit from improved narrative economy, more cautious causal interpretations, and stronger theoretical and pedagogical articulation. A reflection on curricular applicability and the role of educators in assessing motor development would also enhance the discussion’s relevance and depth.

Conclusions

The conclusions appropriately highlight the positive effects of the Grow+ program on motor competence, perceived competence, and perceived physical activity in preschool children. The recognition of hip-hop as a playful yet structured tool for promoting motor development and child health is appreciated. However, the following refinements are recommended:

  • Moderate claims of effectiveness, as the lack of a control group limits causal inference.
  • Avoid broad generalizations such as “the program has proven effective” without acknowledging methodological limitations.
  • Reconsider the use of the term “motor illiteracy” to describe parents and educators. While the concept is provocative, it may be overly categorical without qualitative evidence to support it.
  • Strengthen the connection to educational curricula, emphasizing how such interventions can be integrated into real pedagogical proposals.

References

It is strongly recommended to review the journal’s reference formatting guidelines and ensure full compliance (MDPI reference style: https://www.mdpi.com/authors/references ). The current reference list lacks consistency and contains several formal issues that affect academic presentation:

  • Author names are inconsistently formatted (e.g., full names vs. initials).
  • Some entries include unnecessary translations of titles (e.g., reference 2).
  • There are inconsistencies in capitalization (title case vs. sentence case) and in the inclusion of metadata such as volume, issue, and page numbers.
  • DOIs should be presented as full URLs (e.g., https://doi.org/...), and this should be applied uniformly.

It is advisable to use a reference manager and carefully verify all entries to ensure accuracy, consistency, and adherence to the required citation style.

Author Response

Reviewer 1

 The following comments are intended to provide constructive feedback to enhance the scientific rigor, conceptual clarity, and educational relevance of the manuscript. The observations address key aspects of the study—including its theoretical framing, methodological design, interpretation of results, and formal presentation—with the aim of supporting the authors in refining their work for publication. Each point is offered with the intention of strengthening the manuscript’s contribution to the field of early childhood motor development and educational intervention research.

Response: Dear reviewer, we greatly appreciate your time and dedication in reviewing our manuscript.

We will consider all your comments and suggestions for improvement. In order to make our responses and changes clearer, we will respond here and change the content of the manuscript by highlighting it with a blue background.

General Observations

The sample size (n = 37) is relatively small, which limits the generalizability of the findings. While this limitation can be addressed in the main body of the article, it would be advisable to briefly acknowledge it in the abstract.

Response: Dear reviewer, thank you for your suggestion. In accordance with it, we have added a sentence to the summary emphasizing the need to interpret the data with caution, limiting its generalization.

Additionally, the study does not clarify whether the order of the intervention phases was randomized or whether any external variables were controlled during the rest period.

Response: We thank you for the pertinent comment. The present study followed a quasi-experimental, self-controlled design, in which the order of the intervention and rest phases was fixed and applied equally to all participants (i.e., not randomized). During the rest period, no structured physical activity program was implemented; children continued with their regular preschool routines. To clarify the design and address this concern, we have revised subsection 2.1 (Study Design) in the Materials and Methods and added a schematic representation (now Figure 1) illustrating the sequence of assessments (O) and intervention periods (X). We believe that this table will help readers to better understand our study design.

Furthermore, the decision to assess motor competence (MC), perceived motor competence (PMC), perceived motor coordination (PMCoor), and physical activity (PA) in the context of a hip-hop intervention for early childhood education may appear self-evident. This raises the question of whether the intervention functions more as a pedagogical innovation or as a tool with broader educational implications. Would similar outcomes have emerged from programs based on regional dances, ballet, or other movement forms?

In this regard, while the use of dance is justified, the specific choice of hip-hop is not sufficiently supported—particularly in the introduction and discussion. The rationale for selecting hip-hop over other dance modalities should be clearly articulated, especially considering the developmental stage of the participants.

Response: We acknowledge your observation. While dance in general is indeed a suitable tool for motor development, hip-hop was specifically chosen due to its accessibility, cultural relevance, and adaptability to preschool-aged children. Unlike more technical forms such as ballet, hip-hop involves simple, rhythmic, and inclusive movement patterns that match the coordinative capacities of children aged 3–4. Furthermore, existing literature highlights hip-hop’s motivational value and its effectiveness in promoting engagement, motor skill development, and psychosocial outcomes. We have revised both the introduction and discussion to better articulate these reasons, adding also some more literature.

We agree with the reviewer that the intervention can be understood both as a pedagogical innovation and as a broader educational tool. The Grow+ program not only introduces hip-hop as a novel movement experience within preschool education, but also promotes motor, psychosocial, and health-related outcomes, thus extending its implications beyond pedagogy. We have emphasized this dual role in the revised discussion.

The manuscript uses the terms early childhood and preschool interchangeably. Greater terminological precision is recommended. The lack of clear distinction between these terms may lead to misinterpretations regarding the scope and applicability of the findings. It is suggested that the authors define the chosen term explicitly and justify its use in relation to the study design, ensuring conceptual coherence and population specificity. In particular, the use of early childhood is encouraged, as it frames the intervention within a broader developmental perspective. This choice supports a holistic understanding of motor learning and aligns with international educational frameworks focused on comprehensive early development.

Response: We thank the reviewer for their observation and suggestion. We recognise that the interchangeable use of the terms early childhood and preschool can create ambiguity. In line with their suggestion, we have revised the manuscript to adopt the term early childhood consistently, except when we refer explicitly to the institutional context of the intervention (i.e., the preschool environment). In addition, we have added a brief clarification in the introduction, after the second paragraph, defining early childhood as a broader developmental stage (from birth to 8 years of age) and specifying that the participants in this study were between 3 and 4 years of age and enrolled in preschool education. This adjustment ensures conceptual accuracy, aligns with international education and development frameworks, and supports a holistic understanding of motor learning during this critical period.

Title

The title could benefit from including a reference to the study design or type of intervention, and the name of the program might be more appropriately placed in the subtitle. Additionally, it would be helpful to specify the geographical or educational context of the intervention.

Response: Dear reviewer, thank you for your suggestion. As suggested, we have added the study design and geographical context to the title.

Abstract

The abstract would benefit from improved conceptual clarity and terminological precision:

  • Both perceived motor competence (PMC) and perceived motor coordination (PMCoor) are mentioned, but the distinction between these constructs is not clearly explained. A brief conceptual clarification or justification for evaluating them separately would be helpful.
  • The term self-controlled may be misleading. If the study follows a pre-post design without a control group, it would be clearer to describe it as a within-subjects quasi-experimental design.
  • Response: We have made the suggested change.
  • The phrase positive trend is vague. Does it refer to a non-significant improvement or a linear progression? Greater specificity is recommended.
  • Response: Thank you for the suggestion. Indeed, the sentence could be vague. We would like to clarify that this is an insignificant increase.

Keywords

Keywords should align with established thesauri and avoid redundancy with terms already present in the title or abstract. More concise and specific alternatives are suggested:

  • hip-hop dance, early childhood, motor skills, physical activity, intervention study

Response: Dear reviewer, thank you for your suggestion. We have changed the keywords to avoid redundancy.

Introduction

The use of Stodden’s theoretical model provides a solid and relevant framework for analyzing the constructs under study. Similarly, the justification for using dance as a pedagogical and motor tool is appropriate, including its neurological and psychomotor benefits.

However, the focus on hip-hop requires further elaboration. While general benefits of hip-hop are mentioned, the rationale for selecting this specific modality over others (e.g., ballet, traditional dances, creative movement) for 3–4-year-old children is not sufficiently developed. Key questions remain:

  • What characteristics make hip-hop particularly suitable or effective at this developmental stage?

Response: Dear reviewer, thank you very much for your comment and question. As we explained above in the general comments, while dance in general is indeed a suitable tool for motor development, hip-hop was specifically chosen due to its accessibility, cultural relevance, and adaptability to preschool-aged children. Unlike more technical forms such as ballet, hip-hop involves simple, rhythmic, and inclusive movement patterns that match the coordinative capacities of children aged 3–4. Furthermore, existing literature highlights hip-hop’s motivational value and its effectiveness in promoting engagement, motor skill development, and psychosocial outcomes. We have revised both the introduction and discussion to better articulate these reasons, adding also some more literature.

  • Has the cultural, cognitive, or motor appropriateness of hip-hop for this age group been considered?

Response: Your question is indeed very pertinent. The Grow+ programme is a scientifically validated intervention programme using the Credci2 and CERT methodologies, and its development and validation are explained in a scientific article. During this development and validation process, its cultural, cognitive and motor suitability for this age group was taken into account. Nevertheless, we recognise the relevance of your concern and, in line with this, we have added this clarification in the 1st paragraph oh the “Intervention Program” section.

Bernardino, S.; Saramago, N.; Catela, D.; Branco, M.; Mercê, C. Desarrollo y validación de un programa de intervención hip hop para niños en edad preescolar: Crescer+ (Crecer+) (Development and validation of a hip hop intervention programme for pre-school children: Crescer+ (Grow+)). Retos 2024, 55, 212-225, doi:10.47197/retos.v55.103575.

  • Could the use of hip-hop be more closely linked to pedagogical innovation than to a solid educational rationale?

Response: Thank you for your question, which has prompted us to reflect more deeply on this issue. The Grow+ can be understood both as a pedagogical innovation and as a broader educational tool. The Grow+ program not only introduces hip-hop as a novel movement experience within preschool education, but also promotes motor, psychosocial, and health-related outcomes, thus extending its implications beyond pedagogy. We have emphasized this dual role in the revised discussion.

Moreover, the introduction does not address how this intervention aligns with the early childhood education curriculum or what specific educational needs it aims to meet. This is essential for contextualizing the intervention and assessing its transferability to other socio-educational settings:

  • What current barriers exist to motor competence development in early childhood education?
  • Response:  In Portugal, there is not operational physical education program, only curricular orientations. However, this program is in accordance to the following objective: Master movements that involve movement and balance such as: climbing, running, jumping, sliding, spinning, jumping with feet together or on one foot, jumping over obstacles, swinging, crawling and rolling. (OCEPE, 2016, p. 46). Considering also that, as you mentioned, this information is essential to contextualise the intervention and assess its transferability to other socio-educational contexts, we have added two paragraphs to the introduction that clarify this issue.
  • What role do educators and the school environment play in implementing such programs?
  • Response: Dear reviewer, thank you for your question. In Portugal, private kindergartens hire physical education teachers or exercise technicians to teach physical education classes; in the public school system, both kindergarten teachers and physical education teachers may be involved in implementing physical education classes, depending on the teacher management system for each school group. The production of support materials, involved in this project, enables kindergarten teachers, physical education teachers, and exercise coaches to implement it. Once again, recognising the value of your comment, we have added this information and clarification to the introduction.

The transition from general dance to hip-hop is somewhat abrupt. A smoother transition is recommended to explain how the discussion moves from a broad category (dance) to a specific form (hip-hop). Additionally, the literature review on hip-hop is extensive but lacks critical depth. While many benefits are cited, potential limitations—such as lack of standardization, variability in implementation, or limited evidence in preschool populations—are not addressed.

Response: Dear reviewer, we recognize that the transition from dance in general to hip hop was somewhat abrupt, so we have added another paragraph to the introduction that makes this connection in a more fluid and guided way, moving from the general to the specific. We have also added more literature to the introduction that allows us to understand the specificities and benefits of hip hop. In accordance with your comment, we have also explained the limitations of existing studies, adding a few sentences to the last paragraph of the introduction.

A simple search using “early childhood” AND “hip-hop” in Web of Science reveals more research than is suggested in the introduction. It is therefore recommended to revise the literature to ensure that a significant portion of the references focus specifically on the 3–5 age range. Studies that critically examine the use of hip-hop in educational contexts—not just as a health or entertainment tool—should be included. Furthermore, references to international normative and pedagogical frameworks supporting the inclusion of rhythmic movement in early childhood education would strengthen the argument.

Response: Dear reviewer, thank you very much for your pertinent comments, which made us reflect more deeply and better adjust the text. Following this comment, we added to the introduction (in the paragraph beginning with ‘Although multiple dance modalities’...) literature that focuses specifically on the 3–5 age range and critically examines the use of hip-hop in educational contexts, emphasising its benefits as an educational tool. Also, as suggested, we added references to international normative and pedagogical frameworks supporting the inclusion of rhythmic movement in early childhood education to the last paragraph of the introduction to reinforce the relevance of the topic and study.

Methods

This is the most robust section of the manuscript. It is clearly structured and provides detailed descriptions of the study design, instruments, and evaluation conditions. The use of repeated measures ANOVA and Spearman correlations is appropriate.

However, the section could benefit from improved narrative economy. The level of detail, while thorough, may be excessive for a specialized reader. For instance, the item-by-item description of questionnaires and motor tests could be summarized or moved to an appendix. The essential information for replicability should be retained, but redundancies should be avoided.

Response: Thank you very much for your comment and for acknowledging the robustness of our methods section. We agree that, for a specialized reader, the detailed description of the methods may seem excessive or dense. However, our intention in providing a summary that highlights the essential points (e.g., which questions from the Perceived Physical Activity Level questionnaire or which tests from the motor skills battery were used) is to make the article accessible to readers from diverse backgrounds, not only specialists. For instance, we want parents or early childhood educators to be able to read our article and, even if they are unfamiliar with the specific assessment protocols, still grasp their essence and better understand the results and their implications. For this reason, we have carefully avoided redundancies, as you suggest, while retaining the description of the protocols.

The description of the design as self-controlled is not accompanied by a justification for the absence of a control group. It would be helpful to explain whether logistical, ethical, or contextual constraints prevented its inclusion. Additionally, the potential influence of natural maturation effects in 3–4-year-old children should be briefly discussed.

Response: We thank the reviewer for this important observation. Quasi-experimental designs without a parallel control group are considered a valid methodological option when contextual, ethical, or logistical constraints prevent the allocation of participants to control conditions (Creswell & Creswell, 2018). In our case, it was not possible to establish a control group because the preschool had a single class of 3–4-year-old children, and the school management did not allow the division of children into different conditions. To mitigate this limitation, we adopted a self-controlled design with repeated measures and an internal non-intervention period (i.e., interrupted time series design), which allowed us to compare changes across moments with and without exposure to the program. This approach has been recommended in quasi-experimental research as it enables the identification of intervention effects while reducing some threats to internal validity. Future studies should seek to include randomized control groups or matched comparison groups to further strengthen causal inferences.

Reference: Creswell, J. W., & Creswell, J. D. (2018). Research design: Qualitative, quantitative, and mixed methods approaches (5th ed.). Thousand Oaks, CA: Sage.

Finally, the study evaluates multiple dimensions (MC, PMC, PMCoor, PA, BMI), which enriches the analysis but may dilute the focus. Are all these variables directly aligned with the objectives of the hip-hop program? For example, is the inclusion of BMI justified given the short duration of the intervention? If so, a brief rationale would be appropriate.

Response: Your comment is indeed pertinent. In the introduction, we state that the aim of the study is to "the present study aims to analyze the effects of the 'Grow+' hip-hop program on children in early childhood’s MC, PMC and PA, as well as PMCoor”. In these objectives, BC is not clearly identified, precisely for the reason you mention, i.e. the short duration of the programme. Nevertheless, we present the “Stodden model” as a theoretical and conceptual model, in which BC is one of the central variables and influences the others we wish to evaluate, MC and PA. Furthermore, in the literature review we present, dance intervention studies evaluate and report changes in BC. Thus, recognising the influence of CB on the other variables of interest, as well as in the previous literature, we chose to evaluate it.

Discussion

The discussion offers a detailed interpretation of the results, with attempts to connect them to theoretical and pedagogical frameworks. The effort to analyze the data from multiple dimensions—motor, perceptual, social, and educational—is commendable, as is the intention to project the program’s impact beyond the classroom.

Response: Dear reviewer, we greatly appreciate your words of recognition.

However, several aspects warrant further refinement:

  1. Causal attributions should be moderated.
    The discussion tends to attribute observed changes directly to the intervention without sufficiently considering alternative explanations such as natural maturation, school context, or the Hawthorne effect. Given the absence of a control group, causal claims should be carefully qualified.

Response: We acknowledge that the absence of a control group and the small sample size limit the assessment of causality and generalisation of the results. In this regard, we have reviewed the entire discussion and adapted several excerpts in which the causal relationship and generalisation are clarified and addressed very carefully.

  1. Theoretical integration could be deepened.
    Although Stodden’s model is referenced, the discussion does not elaborate on how hip-hop specifically fits within this framework. It would be valuable to explore mechanisms such as multisensory stimulation, rhythm, and expressive movement. Including references on dance in early childhood education and the role of rhythmic movement in psychomotor development would strengthen the theoretical grounding.

Response: Thank you for the suggestion. In this regard, we have added another paragraph to the discussion that explores the application of Stodden's model in interpreting the results, namely through the mechanisms of multisensory stimulation.

  1. Curricular alignment is underdeveloped.
    While the program is said to promote health and well-being, the discussion does not address how it aligns with early childhood education curricula or what specific educational competencies it supports. A reflection on the feasibility of implementing such programs in real-world school settings is recommended.

Response: Answer: Thank you for your comment and suggestion. We recognise that this topic is relevant and pertinent to readers. As we did in the introduction, we have now discussed the topic in the discussion section, adding the last paragraph to the discussion.

  1. Adult perceptions are underexplored.
    The divergence between educators’ and caregivers’ perceptions is intriguing but not sufficiently unpacked. What does it mean that educators did not perceive improvements? How might teachers be more actively involved in the intervention? These questions have important implications for the educational value and sustainability of the program.

Response: Thank you for the suggestion, which led us to reflect more deeply on these issues. It is possible that educators experienced the ‘anchor effect’, where educators’ initial impressions of the children’s motor abilities served as a cognitive reference point, influencing subsequent evaluations regardless of actual progress.  In fact, the active participation of educators in future editions of the programme could combat this misperception and even contribute to more significant positive results. We have added a reflection on all these issues at the end of the last paragraph, which discusses the perception of educators.

In summary, the discussion would benefit from improved narrative economy, more cautious causal interpretations, and stronger theoretical and pedagogical articulation. A reflection on curricular applicability and the role of educators in assessing motor development would also enhance the discussion’s relevance and depth.

Response: Dear reviewer, we greatly appreciate all your suggestions for the Discussion section, which we have addressed and responded to specifically in the comments above.

Conclusions

The conclusions appropriately highlight the positive effects of the Grow+ program on motor competence, perceived competence, and perceived physical activity in preschool children. The recognition of hip-hop as a playful yet structured tool for promoting motor development and child health is appreciated. However, the following refinements are recommended:

  • Moderate claims of effectiveness, as the lack of a control group limits causal inference.
  • Avoid broad generalizations such as “the program has proven effective” without acknowledging methodological limitations.

Response: We avoided generalisations, clearly and unequivocally acknowledging in the conclusion that "Nevertheless, given the quasi-experimental design and absence of a control group, these findings should be interpreted with caution.

  • Reconsider the use of the term “motor illiteracy” to describe parents and educators. While the concept is provocative, it may be overly categorical without qualitative evidence to support it.

Response: We recognise that the term ‘motor illiteracy’ may be considered provocative and excessive, so we have rewritten the sentence in a more pedagogical way.

  • Strengthen the connection to educational curricula, emphasizing how such interventions can be integrated into real pedagogical proposals.

 Response: As suggested, we added this topic to the conclusion; however, since it was not the main focus of the study, we did not explore it in depth in this section.

References

It is strongly recommended to review the journal’s reference formatting guidelines and ensure full compliance (MDPI reference style: https://www.mdpi.com/authors/references ). The current reference list lacks consistency and contains several formal issues that affect academic presentation:

  • Author names are inconsistently formatted (e.g., full names vs. initials).
  • Some entries include unnecessary translations of titles (e.g., reference 2).
  • There are inconsistencies in capitalization (title case vs. sentence case) and in the inclusion of metadata such as volume, issue, and page numbers.
  • DOIs should be presented as full URLs (e.g., https://doi.org/...), and this should be applied uniformly.

It is advisable to use a reference manager and carefully verify all entries to ensure accuracy, consistency, and adherence to the required citation style.

Response: Thank you kindly for your suggestion. In response, we have chosen to use EndNote software to format the references in accordance with MDPI’s guidelines. We truly appreciate your attention to detail and your support in helping us improve the manuscript.

Reviewer 2 Report

Comments and Suggestions for Authors

General Comments

This manuscript presents a quasi-experimental study investigating the impact of a hip-hop intervention program on motor competence and related variables in preschool children. While the research topic is certainly relevant and timely, I have several significant methodological and analytical concerns that unfortunately limit how we can interpret and generalize the findings.

My most critical concern is the complete absence of a control group. This fundamentally undermines the ability to confidently establish a causal link between the intervention and any observed changes. The authors used a self-controlled design, which, while pragmatic, simply cannot adequately differentiate between the effects of the intervention and the natural developmental progression that occurs in this age group. This is particularly problematic because motor competence naturally improves with age and practice in preschool children.

Beyond the design, the statistical analysis also presents multiple issues. These include inadequate handling of multiple comparisons, some questionable interpretations of non-significant trends, and a problematic use of age as a covariate. Furthermore, the reliance on subjective measures, especially pictorial scales administered to very young children, raises serious questions about the validity and reliability of the measurements for this population. The apparent disconnect between reported objective motor competence improvements and unchanged adult perceptions of those improvements suggests either measurement issues or that any improvements might be specific to the intervention context rather than indicative of generalizable motor development. Lastly, I found that the discussion section often contains interpretations that stretch beyond what the data can reasonably support.

Minor Weaknesses

The manuscript could benefit from being more concise in several sections, particularly the introduction and discussion. Some of the statistical procedures need clearer justification , and the intervention description lacks enough detail for someone else to replicate it. While the literature review is comprehensive, it doesn't quite establish a specific need for hip-hop interventions in this population beyond the general benefits of physical activity.

Specific Comments by Section

Title and Abstract

Page 1, Lines 2-3: The title makes a strong claim about "enhancing" motor competence. Without adequate control conditions, this causal claim feels overstated.

Page 1, Lines 27-28: The abstract mentions "significant improvements in MC were observed during intervention periods" but crucially omits the lack of a control group. This omission is vital for readers to properly contextualize and interpret these claims.

Introduction

Page 2, Lines 89-92: The statement that "only one hip-hop-based intervention program has been applied to preschool children" needs verification. It appears to be based on an incomplete literature search rather than a comprehensive systematic review.

Page 3, Lines 97-101: The study aims are quite broad and lack specific, testable hypotheses. The research questions would benefit from being more precisely formulated to effectively guide the analysis.

Methods

Page 3, Lines 103-110: The study design description fails to explicitly acknowledge the fundamental limitation of not having a control group. This should be directly stated as a limitation, rather than merely implied.

Page 3, Lines 112-119: There's no mention of a power analysis or sample size calculation. For a study with 37 participants across four time points, this is essential to determine if the sample size is sufficient to detect meaningful effects.

Page 3, Lines 127-135: The intervention description is missing crucial details, such as content progression, instructor training protocols, and how fidelity was monitored. These details are vital for both replication and assessing the quality of the intervention.

Page 4, Lines 136-150: While the choreography selection criteria are described, the actual motor skills involved and their developmental appropriateness aren't clearly specified or justified using motor development literature.

Statistical Analysis

Page 6, Lines 237-243: The statistical approach involves multiple comparisons across four time points, but without appropriate correction (e.g., Bonferroni correction). This inflates the risk of a Type I error.

Page 7, Lines 284-287: Using decimal age as a covariate is problematic. The intervention itself might affect children of different ages in different ways, potentially masking important interaction effects.

Methodological Concerns

The absence of intervention fidelity measures raises questions about whether the intervention was delivered consistently across sessions and to all children.

Additionally, the potential for expectancy effects in both the children and the assessors (who appear to have been unblinded) might have influenced the outcomes.

My biggest concern here is the reliance on pictorial scales for measuring perceived competence in 3-4 year-olds. Children this young often have limited metacognitive awareness of their own abilities and may be more influenced by very recent experiences rather than providing stable, reliable assessments.

Results

Page 8, Lines 292-298: The interpretation that "changes in motor competence over time were not solely explained by chronological age" is overstated. Without a control group, it's simply unclear whether the intervention provided benefits beyond what would naturally occur through development.

Page 8, Lines 304-312: The claim of "no significant differences between second and third evaluations" during the break period is interpreted as evidence of an intervention effect. However, this could just as easily represent a ceiling effect or limitations in the measurement itself.

Page 9, Lines 325-331: The interpretation that younger children "developed a perception of motor competence comparable to that of their older peers" lacks statistical support and may well reflect measurement error rather than a meaningful change.

Discussion

Page 13, Lines 394-409: The discussion of a "motor catch-up effect" is highly speculative and isn't supported by the study design. Without a control group, these kinds of developmental interpretations cannot be substantiated.

Page 14, Lines 426-438: The statement that "the difficulty of the motor schemes was well adjusted to the coordinative-motor developmental stage" isn't supported by any objective assessment of task difficulty or developmental appropriateness.

Page 14, Lines 453-472: The discussion of perceived motor competence changes includes interpretations that go beyond what the data can support, especially concerning differential effects on younger versus older children.

Conclusions

Page 16, Lines 532-537: The conclusion that improvements were "not solely explained by age" and demonstrated "program's effectiveness in promoting motor development, independent of chronological age" is simply not justified by the study design. This represents a significant overinterpretation of the findings.

Author Response

Reviewer 2

Response: Dear reviewer, we greatly appreciate your time and dedication in reviewing our manuscript. We will consider all your comments and suggestions for improvement. In order to make our responses and changes clearer, we will respond here and change the content of the manuscript by highlighting it with a green background.

General Comments

 This manuscript presents a quasi-experimental study investigating the impact of a hip-hop intervention program on motor competence and related variables in preschool children. While the research topic is certainly relevant and timely, I have several significant methodological and analytical concerns that unfortunately limit how we can interpret and generalize the findings.

My most critical concern is the complete absence of a control group. This fundamentally undermines the ability to confidently establish a causal link between the intervention and any observed changes. The authors used a self-controlled design, which, while pragmatic, simply cannot adequately differentiate between the effects of the intervention and the natural developmental progression that occurs in this age group. This is particularly problematic because motor competence naturally improves with age and practice in preschool children.

Beyond the design, the statistical analysis also presents multiple issues. These include inadequate handling of multiple comparisons, some questionable interpretations of non-significant trends, and a problematic use of age as a covariate. Furthermore, the reliance on subjective measures, especially pictorial scales administered to very young children, raises serious questions about the validity and reliability of the measurements for this population. The apparent disconnect between reported objective motor competence improvements and unchanged adult perceptions of those improvements suggests either measurement issues or that any improvements might be specific to the intervention context rather than indicative of generalizable motor development. Lastly, I found that the discussion section often contains interpretations that stretch beyond what the data can reasonably support.

Minor Weaknesses

The manuscript could benefit from being more concise in several sections, particularly the introduction and discussion. Some of the statistical procedures need clearer justification, and the intervention description lacks enough detail for someone else to replicate it. While the literature review is comprehensive, it doesn't quite establish a specific need for hip-hop interventions in this population beyond the general benefits of physical activity.

Response: Dear reviewer, we would like to sincerely thank you for the time, care, and dedication you devoted to reviewing our manuscript. Your thoughtful and detailed feedback has been truly appreciated. We carefully considered all of your general comments, which we found to be clearly reflected in the specific points raised throughout the review. In light of this, we will respond to each of the specific comments individually in the sections below, ensuring that your suggestions are addressed with the attention they deserve.

Specific Comments by Section

Title and Abstract

 Page 1, Lines 2-3: The title makes a strong claim about "enhancing" motor competence. Without adequate control conditions, this causal claim feels overstated.

Response: Dear reviewer, thank you very much for your comment. We have replaced the term “enhancing” with “improve”. In line with Reviewer 1's suggestions, we have also reworded the entire title.

Page 1, Lines 27-28: The abstract mentions "significant improvements in MC were observed during intervention periods" but crucially omits the lack of a control group. This omission is vital for readers to properly contextualize and interpret these claims.

Response: We thank the reviewer for this important observation. Quasi-experimental designs without a parallel control group are considered a valid methodological option when contextual, ethical, or logistical constraints prevent the allocation of participants to control conditions (Creswell & Creswell, 2018). In our case, it was not possible to establish a control group because the preschool had a single class of 3–4-year-old children, and the school management did not allow the division of children into different conditions. To mitigate this limitation, we adopted a self-controlled design with repeated measures and an internal non-intervention period, which allowed us to compare changes across moments with and without exposure to the program. This approach has been recommended in quasi-experimental research as it enables the identification of intervention effects while reducing some threats to internal validity. Nevertheless, we acknowledge that natural maturation processes in children aged 3–4 years may have influenced developmental outcomes, and this should be considered when interpreting the findings. Future studies should seek to include randomized control groups or matched comparison groups to further strengthen causal inferences.

Reference: Creswell, J. W., & Creswell, J. D. (2018). Research design: Qualitative, quantitative, and mixed methods approaches (5th ed.). Thousand Oaks, CA: Sage.

Introduction

Page 2, Lines 89-92: The statement that "only one hip-hop-based intervention program has been applied to preschool children" needs verification. It appears to be based on an incomplete literature search rather than a comprehensive systematic review.

Response: Thank you for your comment. We believe that we conducted a comprehensive search, but although there are several hip hop programmes implemented in schools, we only found one programme for the relevant pre-school age group, which was implemented several times and resulted in two scientific articles that we discussed in our introduction. Therefore, we believe that the reason for your comment is not related to limited research, but rather to research specific to a very young age group, for which there is little literature. In order to make this issue clearer to the reader, we have revised that section of the text: “However, according to our review of the literature, despite the existence of several hip-hop-based intervention programmes, only one has been specifically applied to children in early childhood”.

Page 3, Lines 97-101: The study aims are quite broad and lack specific, testable hypotheses. The research questions would benefit from being more precisely formulated to effectively guide the analysis.

Response: Dear reviewer, thank you for your comment. As suggested, we have added the hypotheses at the end of the introduction.

Methods

Page 3, Lines 103-110: The study design description fails to explicitly acknowledge the fundamental limitation of not having a control group. This should be directly stated as a limitation, rather than merely implied.

Response: Dear reviewer, thank you for your comment. As suggested, we have added to the ‘Study design’ section an explanation that this type of study has limitations, and we have also explained the reasons for choosing it, supported by a reference. We believe that thanks to your comment and, consequently, our addition of this information, this section has become clearer and more complete.

Page 3, Lines 112-119: There's no mention of a power analysis or sample size calculation. For a study with 37 participants across four time points, this is essential to determine if the sample size is sufficient to detect meaningful effects.

Response: Dear Reviewer, thank you for your valuable observation. We have now included a priori power analysis in the Methods section to clarify the adequacy of our sample size. The analysis was conducted using G*Power 3.1.9.4 (Faul et al., 2007) for a repeated-measures ANOVA (within-subjects, four time points).

Parameters were set as follows:
Effect size f = 0.25, representing a medium effect as recommended by Cohen (1988). This choice was conservative given the absence of previous studies employing the same intervention and ensures the study was sufficiently powered to detect effects of practical relevance.
α = 0.05 (standard criterion to minimize Type I error).
Power (1–β) = 0.95, a stringent criterion adopted to reduce the probability of Type II error and increase the reliability of detecting real effects.
Number of groups = 1;

Number of measurements = 4.
Correlation among repeated measures (ρ) = 0.50, reflecting a moderate and plausible estimate in line with values typically reported in longitudinal studies with young children.
Nonsphericity correction (ε) = 1.0, assuming sphericity as is standard practice in a priori analyses (Cohen, 1988).

Under these assumptions, the required sample size was 36 participants. Our final sample included 37 children, which slightly exceeded the requirement and thus ensured that the study was adequately powered to detect medium-sized effects. We have added this information to the Methods section of the manuscript.

References:
– Faul, F., Erdfelder, E., Lang, A.-G., & Buchner, A. (2007). G*Power 3: A flexible statistical power analysis program for the social, behavioral, and biomedical sciences. Behavior Research Methods, 39, 175–191.
– Cohen, J. (1988). Statistical power analysis for the behavioral sciences (2nd ed.). Hillsdale, NJ: Lawrence Erlbaum Associates.

Page 3, Lines 127-135: The intervention description is missing crucial details, such as content progression, instructor training protocols, and how fidelity was monitored. These details are vital for both replication and assessing the quality of the intervention.

Response: Dear Reviewer, we appreciate your observation. The intervention (Grow+ hip-hop program) was not newly designed for this study but had already been developed, validated, and published in a previous article (Bernardino et al., 2024). In that paper, we described in detail the program’s theoretical framework, content progression, and developmental rationale. Nevertheless, recognizing the pertinence and importance of your comment, we have expanded the program’s description in the manuscript.

Bernardino, S.; Saramago, N.; Catela, D.; Branco, M.; Mercê, C. Desarrollo y validación de un programa de intervención hip hop para niños en edad preescolar: Crescer+ (Crecer+) (Development and validation of a hip hop intervention programme for pre-school children: Crescer+ (Grow+)). Retos 2024, 55, 212-225, doi:10.47197/retos.v55.103575.

Page 4, Lines 136-150: While the choreography selection criteria are described, the actual motor skills involved and their developmental appropriateness aren't clearly specified or justified using motor development literature.

Response: Thank you for your comment. The content of the intervention was designed in accordance with the literature on motor development and the Curriculum Guidelines for Portuguese Pre-school Education (OCEPE, 2016). Specifically, the choreography incorporated fundamental motor skills (FMS) such as running, jumping, hopping, spinning, and sliding. These motor skills are explicitly recommended by OCEPE and supported by developmental models (e.g., Gallahue et al., 2012; Hulteen et al., 2018). The progression of motor skills was also programmed considering the children's motor development stage, starting with simpler locomotor actions and advancing to integrated sequences that combined stability, locomotion, and rhythmic coordination. Recognising that your question could add depth to the manuscript, we have clarified all these aspects in the manuscript and added references to the literature on motor development to justify its developmental appropriateness.

Statistical Analysis

Page 6, Lines 237-243: The statistical approach involves multiple comparisons across four time points, but without appropriate correction (e.g., Bonferroni correction). This inflates the risk of a Type I error.

Response: Dear reviewer, thank you very much for your comment. We did indeed perform the Bonferroni correction, but this should not be explicit in the text, so we have clarified this point.

Page 7, Lines 284-287: Using decimal age as a covariate is problematic. The intervention itself might affect children of different ages in different ways, potentially masking important interaction effects.

Response: We acknowledge and agree that the intervention may have differential effects depending on the age of the children. Age is indeed a relevant factor that can influence how children respond to the programme. With this in mind, and in the interest of transparency, we conducted the ANOVA considering both approaches: (i) treating decimal age without covariation, and (ii) including age as a covariate. Using age as a covariate allows us to control for its potential influence, thereby isolating the effect of the programme itself. Notably, the results indicated significant programme effects in both models—whether age was controlled for or not. Based on this consistency, we opted to present the results without covariates to simplify interpretation. However, your comment made us reflect on how this dual approach might inadvertently confuse readers. Therefore, we have revised the manuscript to present only the analysis without covariates, aiming for greater clarity and focus. Your suggestion also allowed us to remove a table and some text, which we believe will make the results section easier to read and understand for the reader.

Methodological Concerns

The absence of intervention fidelity measures raises questions about whether the intervention was delivered consistently across sessions and to all children.

Response: Thank you for raising this important point. While we did not apply a formal fidelity checklist in every session due to logistical constraints, we sought to ensure intervention consistency through standardized instructor training, detailed session plans, and the use of a validated program previously published (Bernardino et al., 2024). Instructors were supported by a manual and demonstration videos to guide delivery. This constitutes a limitation, and we have now acknowledged it explicitly in the manuscript, suggesting that future studies incorporate systematic fidelity monitoring.

Bernardino, S.; Saramago, N.; Catela, D.; Branco, M.; Mercê, C. Desarrollo y validación de un programa de intervención hip hop para niños en edad preescolar: Crescer+ (Crecer+) (Development and validation of a hip hop intervention programme for pre-school children: Crescer+ (Grow+)). Retos 2024, 55, 212-225, doi:10.47197/retos.v55.103575.

Additionally, the potential for expectancy effects in both the children and the assessors (who appear to have been unblinded) might have influenced the outcomes.

Response: The participating children were blind to the study hypotheses and were not informed of the expected outcomes, which minimizes expectancy effects on their side. However, we acknowledge that the assessors were not blinded, which may introduce potential observer bias. To reduce this risk, standardized protocols were used for all assessments, the instruments employed had objective scoring criteria, and assessors received prior training to ensure consistency in data collection.

In response to the recurring suggestion across several comments to clearly address the study’s limitations, we have added a dedicated paragraph in the discussion section. This new addition outlines and clarifies all relevant limitations of the present manuscript, while also offering suggestions to guide future research.

My biggest concern here is the reliance on pictorial scales for measuring perceived competence in 3-4 year-olds. Children this young often have limited metacognitive awareness of their own abilities and may be more influenced by very recent experiences rather than providing stable, reliable assessments.

Response: We appreciate your insightful comment and fully agree that the use of the scale to assess motor competence in children aged 3 and 4 years presents a methodological limitation. Acknowledging this concern—and in response to your suggestion to make the manuscript more concise—we have decided to remove this variable from the study. At this point, all methods employed to collect the variables of interest are properly validated for children in this age group. Thank you for prompting us to reflect more deeply on this issue.

Results

Page 8, Lines 292-298: The interpretation that "changes in motor competence over time were not solely explained by chronological age" is overstated. Without a control group, it's simply unclear whether the intervention provided benefits beyond what would naturally occur through development.

Response: Dear Reviewer, as kindly mentioned earlier, the experimental design included a break period as a way to address the absence of a control group. This approach is considered a valid methodological alternative, particularly within the framework of interrupted time series designs. We truly appreciate your attention to detail and the opportunity to clarify this aspect of our study.

Page 8, Lines 304-312: The claim of "no significant differences between second and third evaluations" during the break period is interpreted as evidence of an intervention effect. However, this could just as easily represent a ceiling effect or limitations in the measurement itself.

Response: Thank you for raising this important point. We understand your concern regarding the possibility of a ceiling effect or limitations in the measurement tools. Our interpretation, however, is based on the observation that improvements were evident both before and after the break period. This pattern suggests that participants continued to benefit from the intervention, which would be unlikely if a ceiling effect were present.

Page 9, Lines 325-331: The interpretation that younger children "developed a perception of motor competence comparable to that of their older peers" lacks statistical support and may well reflect measurement error rather than a meaningful change.

Response: We understand your concern and, acknowledging that you had previously highlighted the limitations of using this method with this age group, we have decided to remove this variable from the study. As a result, this issue is no longer present in the manuscript.

Discussion

Page 13, Lines 394-409: The discussion of a "motor catch-up effect" is highly speculative and isn't supported by the study design. Without a control group, these kinds of developmental interpretations cannot be substantiated.

Response: We agree with your observation, so we have removed the ‘motor catch-up effect’ from the discussion.

Page 14, Lines 426-438: The statement that "the difficulty of the motor schemes was well adjusted to the coordinative-motor developmental stage" isn't supported by any objective assessment of task difficulty or developmental appropriateness.

Response: Dear reviewer, thank you for your comment. We agree with you; in fact, the sentence is not supported by any objective assessment of task difficulty or developmental appropriateness, so we have removed that part and readjusted it.

Page 14, Lines 453-472: The discussion of perceived motor competence changes includes interpretations that go beyond what the data can support, especially concerning differential effects on younger versus older children.

Response: We understand your concern and, acknowledging that you had previously highlighted the limitations of using this method with this age group, we have decided to remove this variable from the study. As a result, this issue is no longer present in the manuscript.

Conclusions

Page 16, Lines 532-537: The conclusion that improvements were "not solely explained by age" and demonstrated "program's effectiveness in promoting motor development, independent of chronological age" is simply not justified by the study design. This represents a significant over interpretation of the findings.

Response: We appreciate this valuable comment and agree that our previous formulation overstated the findings. Our intention was to note that, although chronological age was statistically considered, improvements were observed specifically during intervention phases. Nevertheless, we acknowledge that due to the quasi-experimental design without a parallel control group, it is not possible to claim that the program’s effectiveness was independent of age. We have therefore revised the conclusion to adopt more cautious language.

Round 2

Reviewer 1 Report

Comments and Suggestions for Authors

I sincerely thank you for your thoughtful and comprehensive responses to the comments. Your detailed clarifications and the revisions made to the manuscript reflect a genuine commitment to academic rigor and pedagogical relevance.

I appreciate the care with which you have addressed each point, from methodological adjustments to theoretical elaborations and curricular alignment. The integration of the suggestions has clearly strengthened the manuscript and enhanced its contribution to the field.

Everything is in order from me side, and i ma pleased with the improvements made.

Author Response

Dear Reviewer,

Thank you sincerely for your generous and thoughtful feedback. We deeply appreciate the time, dedication, and academic rigor you brought to the review process. Your comments not only guided us in refining the manuscript but also encouraged us to reflect more profoundly on its theoretical and methodological foundations.

The improvements achieved are largely due to your valuable contributions, which helped us strengthen the manuscript’s clarity, relevance, and impact. It has been a privilege to benefit from your expertise.

With gratitude,

Best Regards.

Reviewer 2 Report

Comments and Suggestions for Authors

General Comments

My primary feedback relates to clarifying the methodology and softening some of the interpretative language.

While the manuscript has improved, the core limitation is still the quasi-experimental design, which lacks a control group. You acknowledge this limitation well, which is appreciated, but it's important to ensure the language used throughout the manuscript consistently reflects this. Some of the conclusions occasionally read a bit too strongly, implying a level of causality that the design cannot fully support. My suggestion would be to use more cautious and speculative language when discussing the intervention's impact (e.g., "our findings suggest..." or "it is possible the intervention contributed to...").

From a methodological standpoint, there are a few areas that would benefit from more detail to aid in transparency and future replication. For instance, the section on instructor training could be expanded to explain how consistency was maintained across sessions. Likewise, the statistical analysis section would be stronger with more information on how you handled missing data and tested for assumptions beyond sphericity.

Finally, while the writing quality is better, the manuscript would still benefit from a thorough copy-edit. I noticed several persistent grammatical errors (particularly with article usage and subject-verb agreement) that sometimes interrupt the flow of reading. Polishing the language will help ensure your important findings are communicated as clearly as possible.

Specific Comments (by page and line)

Here are some more detailed suggestions:

Page 1, Lines 23-24: The phrase "within-subjects quasi-experimental design study was conducted" feels a little clunky. I’d suggest rephrasing for a smoother read, perhaps to something like: "A quasi-experimental within-subjects design was used."

Page 1, Lines 32-33: The caveat about the small sample size in the abstract feels premature. Without the full context of the methods, it might be confusing for readers. Perhaps this point is better placed in the main body of the paper.

Page 2, Lines 64-69: I found the structure of this paragraph a bit confusing. The parenthetical definition of "early childhood" disrupts the flow. You might consider reorganizing this section to present the ideas more sequentially.

Page 4, Lines 168-180: In the power analysis, the rationale for assuming a correlation of ρ=0.50 wasn't immediately clear. It would be helpful to briefly justify this choice, perhaps by citing literature that supports a similar assumption in this context.

Page 4, Lines 181-182: I spotted a small typo here: "aged between 3 and 4 s". I assume this should be "3 and 4 years."

Page 5, Lines 211-216: Regarding the instructor training, could you provide more detail on the standardization procedures? It wasn't clear how consistency was ensured between sessions, especially without formal fidelity monitoring.

Page 6, Lines 268-270: I noticed that the radar gun specification is mentioned twice in the methods. You can likely remove one of these to avoid redundancy.

Page 8, Lines 342-346: The interpretation that age effects diminished due to the intervention should be phrased more cautiously. Is it possible this could reflect measurement issues or other confounding factors rather than a direct treatment effect? It might be worth acknowledging this alternative.

Page 9, Lines 347-348: For statistical reporting, please ensure you report the exact F-statistic values along with both degrees of freedom. For example: "F(3, XX) = 5.744," where XX is the error degrees of freedom.

Page 9, Lines 362-363: I noticed some inconsistencies in how tables are referenced throughout the manuscript. A quick check for consistency would be great.

Page 13, Lines 451-461: This paragraph seems to overstate the findings. I would strongly recommend toning down the causal language. Using phrases like "it is possible that participation in the intervention contributed to..." would be more appropriate given the study's design.

Page 14, Lines 478-484: The discussion about the results plateauing feels a little oversimplified. Could there be other explanations for the non-significant findings between O3 and O4? Exploring alternative possibilities would add more depth to the discussion.

Author Response

Response: Dear reviewer, we greatly appreciate your time and dedication in reviewing our manuscript. We will consider all your comments and suggestions for improvement. In order to make our responses and changes clearer, we will respond here and change the content of the manuscript by highlighting it with a green background.

General Comments

My primary feedback relates to clarifying the methodology and softening some of the interpretative language.

While the manuscript has improved, the core limitation is still the quasi-experimental design, which lacks a control group. You acknowledge this limitation well, which is appreciated, but it's important to ensure the language used throughout the manuscript consistently reflects this. Some of the conclusions occasionally read a bit too strongly, implying a level of causality that the design cannot fully support. My suggestion would be to use more cautious and speculative language when discussing the intervention's impact (e.g., "our findings suggest..." or "it is possible the intervention contributed to...").

Answer: As suggested, we have carefully reviewed the language throughout the manuscript, avoiding speculative expressions and adopting a more measured approach in the discussion and conclusions. We appreciate your comment.

From a methodological standpoint, there are a few areas that would benefit from more detail to aid in transparency and future replication. For instance, the section on instructor training could be expanded to explain how consistency was maintained across sessions. Likewise, the statistical analysis section would be stronger with more information on how you handled missing data and tested for assumptions beyond sphericity.

 Answer: We believe we have addressed your comment through the responses and adjustments to your various specific comments described below.

Finally, while the writing quality is better, the manuscript would still benefit from a thorough copy-edit. I noticed several persistent grammatical errors (particularly with article usage and subject-verb agreement) that sometimes interrupt the flow of reading. Polishing the language will help ensure your important findings are communicated as clearly as possible.

Answer: Thank you for your suggestion. We have reviewed the grammar and language accordingly.

Specific Comments (by page and line)

 Here are some more detailed suggestions:

Page 1, Lines 23-24: The phrase "within-subjects quasi-experimental design study was conducted" feels a little clunky. I’d suggest rephrasing for a smoother read, perhaps to something like: "A quasi-experimental within-subjects design was used."

Answer: Dear Reviewer, thank you for your suggestion. We have made the change.

Page 1, Lines 32-33: The caveat about the small sample size in the abstract feels premature. Without the full context of the methods, it might be confusing for readers. Perhaps this point is better placed in the main body of the paper.

Answer: Thank you for the suggestion. We have removed that sentence from the abstract.

Page 2, Lines 64-69: I found the structure of this paragraph a bit confusing. The parenthetical definition of "early childhood" disrupts the flow. You might consider reorganizing this section to present the ideas more sequentially.

Answer: Indeed, we found the paragraph somewhat confusing, so we have rewritten it to present the concept and its rationale in a more fluent and sequential manner.

Page 4, Lines 168-180: In the power analysis, the rationale for assuming a correlation of ρ=0.50 wasn't immediately clear. It would be helpful to briefly justify this choice, perhaps by citing literature that supports a similar assumption in this context.

Answer: Thank you for your observation. We have now clarified the rationale for assuming a correlation of ρ=0.50 in the power analysis. This value was selected as a moderate and plausible estimate, commonly used in repeated-measures designs involving young children, especially when prior empirical data are unavailable. According to methodological literature on G*Power and repeated-measures ANOVA, values between 0.30 and 0.60 are typically adopted in similar contexts (e.g., Langenberg et al., 2023). We have added a brief justification to the manuscript to reflect this.

Page 4, Lines 181-182: I spotted a small typo here: "aged between 3 and 4 s". I assume this should be "3 and 4 years."

Answer: Thank you for pointing that out, we have corrected the typo.

Page 5, Lines 211-216: Regarding the instructor training, could you provide more detail on the standardization procedures? It wasn't clear how consistency was ensured between sessions, especially without formal fidelity monitoring.

Answer: Thank you for your comment. We have now expanded the description of the standardization procedures used during instructor training. Although formal fidelity monitoring was not implemented, consistency across sessions was ensured through a combination of structured training, detailed session plans, and the use of a validated instructional manual and video materials. These resources provided clear guidance on choreography progression, motor skill targets, and pedagogical strategies, allowing instructors to deliver the program uniformly. We have clarified this in the revised manuscript.

Page 6, Lines 268-270: I noticed that the radar gun specification is mentioned twice in the methods. You can likely remove one of these to avoid redundancy.

Answer: Thanks for the suggestion, we did just that.

Page 8, Lines 342-346: The interpretation that age effects diminished due to the intervention should be phrased more cautiously. Is it possible this could reflect measurement issues or other confounding factors rather than a direct treatment effect? It might be worth acknowledging this alternative.

Answer: Dear reviewer, thank you for your observation. While we agree that caution is always warranted when interpreting correlational changes, we believe the observed pattern is consistent with the theoretical framework guiding this study. According to Stodden’s model, motor competence in early childhood is highly sensitive to practice opportunities, and structured interventions can accelerate development independently of age. In our sample, younger children reached similar motor competence levels to older peers following the intervention, which suggests that the program may have mitigated age-related disparities. Although measurement error and confounding factors cannot be entirely ruled out, the consistency of the results across multiple assessment points and the alignment with theoretical expectations support the interpretation of a meaningful intervention effect. We have clarified this point in the manuscript.

Page 9, Lines 347-348: For statistical reporting, please ensure you report the exact F-statistic values along with both degrees of freedom. For example: "F(3, XX) = 5.744," where XX is the error degrees of freedom.

Answer: Thanks for the suggestion, we have added the missing information.

Page 9, Lines 362-363: I noticed some inconsistencies in how tables are referenced throughout the manuscript. A quick check for consistency would be great.

Answer: Thank you for pointing that out. We detected two errors in the table identification, which we have corrected.

Page 13, Lines 451-461: This paragraph seems to overstate the findings. I would strongly recommend toning down the causal language. Using phrases like "it is possible that participation in the intervention contributed to..." would be more appropriate given the study's design.

Answer: Thank you, we have taken your suggestion into consideration.

Page 14, Lines 478-484: The discussion about the results plateauing feels a little oversimplified. Could there be other explanations for the non-significant findings between O3 and O4? Exploring alternative possibilities would add more depth to the discussion.

Answer: Thank you for the suggestion, we have added this reflection to the discussion.